# OpenReview forum: "Look Locally, Learn Precisely: Interpretable and Unbiased Text-to-Image Generation with Background Fidelity"
_ICLR.cc/2026/Conference — ICLR 2026 Conference Withdrawn Submission_

### Official Review · Reviewer_35Qb · 2025-10-30

**Soundness:** 3
**Presentation:** 3
**Contribution:** 2
**Rating:** 4
**Confidence:** 5

**Summary:**

This paper proposes a method to achieve unbiased text-to-image generation within the h-space framework of diffusion models, addressing key limitations such as subject misalignment, fairness bias, reduced photorealism, and incoherent backgrounds.
During training, an attribute-separation mask suppresses target attributes in the MCA module, a spatial weighting map removes biased components directly from the h-vector, and a spatially weighted loss focuses optimization on target-relevant regions.
At inference, the low-frequency components of the h-vector are enhanced to improve background fidelity.
The proposed approach retains the interpretability and linear controllability of h-space while enhancing fairness and visual consistency. Experiments on WinoBias, COCO-30k, and I2P datasets with both Stable Diffusion and SDXL show improved background quality and unbiased subject representation compared with previous h-space methods.

**Strengths:**

- Clearly defines the four major weaknesses of prior h-space editing—subject misalignment, fairness limitation, reduced photorealism, and incoherent background—and systematically addresses each.
- Demonstrates high clarity and methodological consistency.
- Evaluated across multiple datasets (WinoBias, COCO-30k, I2P) and diffusion backbones (Stable Diffusion, SDXL), confirming reproducibility and generality.
- Inference method is modular and backward-compatible with existing h-space frameworks, enabling practical integration.
- Offers real-world relevance for fairness-aware or safety-critical image generation tasks.
Improves fairness, photorealism, and controllability simultaneously while preserving interpretability.

**Weaknesses:**

- **Incremental contribution**: The approach is conceptually straightforward and mainly refines prior h-space mechanisms rather than introducing a fundamentally new idea.
- **Limited comparative analysis**: Experiments focus solely on h-space methods; there is no comparison with recent diffusion-based debiasing frameworks such as Unified Concept Editing [1], Fair Diffusion [3], or Entanglement-Free Attention [4].
- **Questionable assumption**: The premise that “backgrounds correspond to low-frequency components” may not generalize to complex or textured scenes (e.g., city, forest, or textured environments).
- **Restricted attribute scope**: Only simple attributes (gender, race, hairstyle) are tested, leaving uncertainty about multi-attribute or context-dependent cases (e.g., age, culture, wedding).
- **Prompt simplicity**: Experiments use short, isolated prompts; performance on complex or narrative prompts is unexplored.
- **Computational cost**: DAAM-based heatmap generation and spatial mask computation add overhead that is not quantified.

**Questions:**

1. Does the proposed low-frequency enhancement remain effective for complex, high-texture backgrounds (e.g., cityscapes or forests)?
2. How would the results differ if FFT or other frequency transforms were used instead of DWT?
3. Could the authors provide specific failure cases or qualitative examples where the method does not work well?
4. When combining distinct concept vectors (e.g., Black person + luxury house), does semantic interference occur?
5. In the spatially weighted loss, how is the identity matrix defined, and does it correctly emphasize target-attribute regions? I guess it should be an all-one matrix rather than an identity matrix.
6. Why is the attribute-separation mask applied only to the final encoder layer instead of multiple layers?
7. Would similar effects be observed if the same training loss were applied to non-h-space diffusion methods?
8. Has the approach been tested on multi-attribute or compositional fairness cases (e.g., elderly Asian woman in a modern office)?
9. Since the attribute-separation mask and spatial weighting map are crucial in training, why are they not reused during inference to further suppress residual bias? How does the inference generate unbiased text-to-image generation without these trained mask and map?


[1] Gandikota, R. et al. “Unified concept editing in diffusion models.” WACV (2024).

[2] Shen, X. et al. “Finetuning Text-to-Image Diffusion Models for Fairness.” ICLR (2024).

[3] Friedrich, F. et al. “Fair diffusion: Instructing text-to-image generation models on fairness.” arXiv preprint arXiv:2302.10893 (2023).

[4] Park, J. et al. “Fair Generation without Unfair Distortions: Debiasing Text-to-Image Generation with Entanglement-Free Attention.” arXiv preprint arXiv:2506.13298 (2025).

[5] Kim, E. et al. “Rethinking Training for De-biasing Text-to-Image Generation: Unlocking the Potential of Stable Diffusion.” CVPR (2025).

---

### Official Review · Reviewer_RAMN · 2025-11-01

**Soundness:** 3
**Presentation:** 2
**Contribution:** 2
**Rating:** 4
**Confidence:** 4

**Summary:**

The paper proposes a training-free framework for interpretable and unbiased text-to-image generation by improving concept learning in the U-Net bottleneck (h-space). It introduces spatially focused attribute suppression during concept vector learning, via attribute-separation masks, spatial weighting, and a spatially weighted loss.

**Strengths:**

1. Spatial weighting map m, derived from inverted DAAM heatmaps, directly attenuates target features in the h-vector, ensuring the concept vector v exclusively captures the attribute.

2. The method is modular and plug-and-play: Fig. 4 shows consistent CLIP gains when applied to PCA-S, H-G, and Self-dis.

3. Results generalize to SDXL, with consistent gains in FID and CLIP (Table 1).

**Weaknesses:**

1. My most concern is the scablity to modern DIT-based models, e.g., Flux. The method is designed for Unet-based methods, can it be extended to modern  models? How to modify the attention block?

2. The attribute-separation mask is applied to “the last MCA module” in Sec. 3.1, but Fig. 7 and Algorithm 2 imply application across all layers, creating inconsistency.

3. Initialization method for concept vector v (e.g., Gaussian, zero) is omitted in Algorithm 2 and Sec. 3.1.

4. No comparison of inference latency with/without low-frequency enhancement is provided (see Sec. 4.2).

5. The overall pipeline seems very complicated (see Figure 2).

**Questions:**

Please see the weakness

---

### Official Review · Reviewer_H8tV · 2025-11-01

**Soundness:** 2
**Presentation:** 2
**Contribution:** 1
**Rating:** 2
**Confidence:** 4

**Summary:**

This paper addresses key limitations of existing text-to-image diffusion models (e.g., subject misalignment, fairness issues, poor background coherence) by manipulating the U-Net’s bottleneck (h-space), which offers interpretability and linear controllability. It proposes two core innovations: (1) A spatially focused concept learning framework that disentangles target attributes (e.g., gender, profession) into concept vectors via three mechanisms: 1. attribute-separation masks (suppressing target features in multi-head cross-attention, MCA), 2. attribute-attentive heatmaps (attenuating target features in h-vectors), and 3. a spatially weighted reconstruction loss (emphasizing attribute-relevant regions). (2) An inference-time low-frequency enhancement strategy that improves background consistency by amplifying low-frequency components in the h-space (via Discrete Wavelet Transform, DWT). Experiments on Stable Diffusion (SD) v1.4 and SDXL using benchmarks like WinoBias (36 subjects), COCO-30k, and I2P (inappropriate content) show the method outperforms state-of-the-art h-space baselines (PCA-S, H-G, Self-dis) in fairness, subject fidelity, background alignment, and image quality.

**Strengths:**

1. **Novel Spatially Targeted Attribute Disentanglement**
   1. The attribute-separation mask ($\chi$) suppresses target features in MCA modules by identifying pixels attending more to conditioning prompts ($\psi$) than target attributes (T).
   2. The spatial weighting mask (m) directly attenuates T in h-vectors via sigmoid-modulated heatmaps.
   3. The spatially weighted loss (Lw) focuses optimization on T-relevant regions.
2. **Modular Inference-Time Background Improvement**
    1. The low-frequency enhancement ($h^\prime = (h+v) + \lambda \overline{h}_{LL}$) is compatible with all h-space methods.
    2. The strategy is parameter-efficient (only $\lambda$ needs tuning).
    3. It maintains foreground quality while enhancing backgrounds.
3. **Comprehensive Empirical Validation Across Use Cases**
   1. Fairness generalization to diverse societal groups.
   2. Safety on inappropriate content.
   3. Scalability to large models..
4. **Interpretability and Controllability**
   1. Linear concept vector weighting.
   2. Composable concept vectors.

**Weaknesses:**

1. **Incomplete Hyperparameter and Generalization Analysis**
   1. Critical hyperparameters lack sensitivity analysis. The paper uses $\tau=0.5$ (MCA mask threshold), $\beta=0.4 $(loss weight), and $\lambda=0.35$ (low-frequency scale) but does not test how these values perform across diverse prompts (e.g., abstract vs. photorealistic) or datasets (e.g., FFHQ). This impacts practical guidance, as users cannot adjust parameters for their use cases.
   2. Lack of statistical significance. Tables 1, 7–8 report average metrics but no error bars or p-values, making it unclear if differences from baselines are statistically meaningful.
2. **Mathematical and Notation Ambiguities**
   1. MCA mask construction lacks formal clarity. Equation 1 defines the normalized margin score ($\delta$) but does not specify how the threshold ($\tau$) is chosen (e.g., validation on a held-out prompt set). Additionally, the paper refers to “attention weights of the final MCA module” (Section 3.1) but does not justify why other layers are excluded. This creates ambiguity for implementers, who cannot confirm if layer choice impacts performance.
   2. DWT/IWT details are missing. Equation 5 references $\mathrm{DWT}(h) = [h_{LL}, h_{LH}, h_{HL}, h_{HH}]$ but does not specify the wavelet type (e.g., Daubechies, Haar) or decomposition level. Appendix C’s pseudo-code (Algorithm 3) also omits these details. This hinders reproducibility, as different wavelets can produce different low-frequency components.
   3. Attribute-attentive heatmap generation is underspecified. The paper uses DAAM (Tang et al. 2023) to generate heatmaps (Section 3.1) but does not detail how DAAM is configured (e.g., attention head selection, heatmap normalization). Appendix B mentions “target attribute–attentive heatmap ˆI=D(T)” but provides no mathematical definition of D(·). This makes it difficult to replicate the heatmap-based mask (m) construction.
3. **Limited Ablation of Core Mechanisms**
   1. No isolation of the spatially weighted loss. Table 9 ablates $\chi$ and m but not the weighted loss (Lw). It is unclear if Lw contributes independently to fairness/fidelity or if its benefits are redundant with $\chi$/m. For example, removing Lw might not impact performance if $\chi$/m already suppress spurious features, making the loss unnecessary.
   2. MCA mask layer choice is untested. The paper applies $\chi$ only to the “final MCA module” (Section 3.1) but does not test applying it to earlier layers or all layers. Appendix B (Figure 7) shows the final layer but no comparisons, leaving uncertainty about optimal layer selection.
   3. No comparison to non-h-space fairness methods. Experiments only compare to h-space baselines (PCA-S, H-G, Self-dis) but not to state-of-the-art fairness methods like ORES (Ni et al. 2024) or DebiasDiffusion (Gandikota et al. 2023). This limits assessment of the method’s standing relative to the broader fairness literature.
4. **Reproducibility and Limitation Usage of Method**
   1. Training/inference time and memory are unreported. Section 4 mentions using an NVIDIA H100 (80 GB) but does not report wall-clock time for concept learning (10k steps) or inference per image. Appendix E notes “1k generated images per concept” but no details on batch processing speed. This limits practical adoption, as users cannot estimate hardware requirements.
   2. Prompt diversity is limited. WinoBias prompts focus on professions (e.g., “doctor”, “nurse”) with simple backgrounds (e.g., “hospital”), but no tests on complex prompts (e.g., “a cyberpunk artist in a neon-lit studio”) or cross-cultural prompts (e.g., non-Western professions). This raises questions about how the method performs on less structured inputs.
   3. The method only applies to text-to-image diffusion models with U-Net architectures (SD and SDXL). At the date of this paper submitted, these two models are already quite outdated and suboptimal models compared to more recent text-to-image models which utilized transformer as architecture and flow matching or AR as generation objective. This limits the method's contribution to the T2I community.

**Questions:**

1. **How do key hyperparameters ($\tau$, $\beta$, $\lambda$) perform across diverse prompts and datasets, and what guidance can be provided for tuning them?** The paper uses fixed values ($\tau$=0.5, $\beta$=0.4, $\lambda$=0.35) but does not test their sensitivity to prompt type (e.g., abstract vs. photorealistic) or dataset (e.g., FFHQ vs. COCO). Could you add a supplementary table showing across $\tau$∈{0.3,0.5,0.7}, $\beta$∈{0.2,0.4,0.6}, and $\lambda$∈{0.2,0.35,0.5} for 3–5 diverse prompts? Additionally, could you provide a heuristic for choosing these parameters (e.g., “increase $\lambda$ for prompts with detailed backgrounds”)?
2. **Can you formalize the MCA mask’s layer selection and DWT configuration, and provide evidence for their optimality?** The paper applies the MCA mask ($\chi$) only to the final layer but does not justify this choice. Could you add an ablation comparing mask application to the final layer, all layers, and early layers (e.g., Layer 1 vs. Layer L) on WinoBias prompts? Additionally, could you specify the DWT wavelet type and decomposition level used in experiments (e.g., Haar wavelet, 2 levels) and test if other configurations impact scores?
3. **How does the method compare to state-of-the-art non-h-space fairness methods (e.g., ORES, DebiasDiffusion) in terms of fairness, quality, and efficiency?** Experiments only compare to h-space baselines, which limits assessment of the paper’s soundness and contribution. Could you add a table comparing your method to ORES (Ni et al. 2024) and DebiasDiffusion (Gandikota et al. 2023) on WinoBias and COCO-30k? Also, report training/inference time for each method to assess efficiency—does the h-space approach offer speed advantages over model fine-tuning methods (e.g., DebiasDiffusion)? Only comparing to h-space baselines is not sufficient for a strong claim of superiority.
4. **Could we general this methods to non-U-Net architectures or more recent text-to-image models?** The method is demonstrated only on U-Net-based diffusion models (only SD v1.4, SDXL), which are outdated and suboptimal in the T2I community. Could you discuss how the spatially targeted disentanglement and low-frequency enhancement strategies could be adapted to some transformer-based architectures in 2025 (or maybe some hybrid models like UViT are welcomed) or models using flow matching objectives?

Overall, the paper presents a technically sound framework for improving text-to-image diffusion models via spatially targeted attribute disentanglement and low-frequency enhancement. However, it falls short of a higher score due to incomplete hyperparameter analysis, mathematical ambiguities (MCA mask construction, DWT details), limited ablations (spatially weighted loss, MCA layer choice), and lack of comparisons to non-h-space fairness methods. More importantly, the method's applicability is limited to U-Net-based diffusion models like SD and SDXL. I tend to reject the paper in its current form but I am willing to hear from authors during rebuttal.

---

### Official Review · Reviewer_UNUT · 2025-11-04

**Soundness:** 2
**Presentation:** 2
**Contribution:** 2
**Rating:** 4
**Confidence:** 5

**Summary:**

This paper aims to add a bit more control over text-to-image (T2I) generation and then use that for fair T2I generation in UNet-based models. For this purpose, they have proposed two major mechanisms:

- **Spatially Focused Concept/ Attribute Learning (Training Phase):**
A method to disentangle and localize concept representations using (a) attribute-separation masks, b) attribute-attentive heatmaps, and c) spatially weighted loss.

- **Low-Frequency Enhancement (Inference Phase):**
They aim to enhance the low-frequency components in the h-space to increase the fidelity of the background generation in the inference time, an aspect which is largely ignored by previous h-space approaches.

**Strengths:**

The high-level idea of focusing on more fine-grained information extraction for a target attribute seems interesting to me, as it provides very good control over T2I generation, although I’m not convinced that the approach they chose to achieve this is generic and transferable.

**Weaknesses:**

I have 3 major concerns regarding this work in general:

$ $

i)  This paper discusses several limitations of previous works (especially h-space approaches), such as:
- *"Existing h-space methods typically learn target attributes from the entire image region Li et al. (2024b); Parihar et al. (2024); Haas et al. (2024), which can entangle them with spurious attributes."* or
- *"Notably, previous h-space methods entirely disregarded both localized learning and inference-time strategies, an oversight that led
to **four major limitations**. One limitation of existing methods is their occasional failure to generate images that align with the
prompt. ... Second, their ability to ensure fairness across different societal groups, ... Third, these approaches often result in poor image quality, ... Fourth, they struggle to accurately generate background content."*

However, these claims are not supported by empirical evidence or other detailed descriptions. Since these points form the primary motivation for this study, the authors need to provide more concrete details and supporting analysis.

$ $

ii) My main concern about this work is that it appears to be specific to models that use a U-Net architecture in their diffusion structure. The proposed idea does not seem generalizable to other models—particularly recent state-of-the-art open-source T2I models such as Stable Diffusion 3 and Qwen-Image, which do not rely on U-Net architectures in order to scale to larger datasets and model sizes.

$ $

iii) This paper also overlooks an important line of recent research on fair T2I image generation via prompt learning (e.g., [1] and [2]). These prompt learning approaches are model-agnostic and more efficient, as they learn only a limited number of tokens for each attribute and can be applied across different T2I models once trained.

$ $

I am willing to increase my score if the conrens are addressed properly.

$ $

*References:*

[1] ITI-GEN: Inclusive Text-to-Image Generation (ICCV'23, Oral)

[2] FairQueue: Rethinking Prompt Learning for Fair Text-to-Image Generation (NeurIPS'24)

**Questions:**

Please check the weaknesses

---

### Note · Authors · 2025-11-13

**Comment:**

We thank the reviewers for their constructive comments, and after careful assessment we have reached the decision to withdraw this submission.

**Withdrawal Confirmation:**

I have read and agree with the venue's withdrawal policy on behalf of myself and my co-authors.